# *Lactiplantibacillus plantarum* LPJZ-658 Improves Non-Alcoholic Steatohepatitis by Modulating Bile Acid Metabolism and Gut Microbiota in Mice

**DOI:** 10.3390/ijms241813997

**Published:** 2023-09-12

**Authors:** Liming Liu, Liquan Deng, Wei Wei, Chunhua Li, Yuting Lu, Jieying Bai, Letian Li, Heping Zhang, Ningyi Jin, Chang Li, Cuiqing Zhao

**Affiliations:** 1College of Animal Science and Technology, Jilin Agricultural Science and Technology University, Jilin 132101, China; aliuliming1984@126.com (L.L.); chunhuali2009@sina.com (C.L.); luyuting2018@163.com (Y.L.); 2School of Public Health, Jilin University, Changchun 130021, China; dengliquan@126.com; 3State Key Laboratory of Pathogen and Biosecurity, Beijing Institute of Microbiology and Epidemiology, Beijing 100071, China; weiwei_7776@hotmail.com; 4College of Future Technology, Peking University, Beijing 100871, China; jieying.bai@pku.edu.cn; 5Research Unit of Key Technologies for Prevention and Control of Virus Zoonoses, Chinese Academy of Medical Sciences, Changchun Veterinary Research Institute, Chinese Academy of Agricultural Sciences, Changchun 130122, China; letian823@163.com (L.L.); ningyik@126.com (N.J.); 6Key Laboratory of Dairy Biotechnology and Engineering, Ministry of Education, Inner Mongolia Agricultural University, Hohhot 010018, China; hepingdd@vip.sina.com

**Keywords:** *Lactiplantibacillus plantarum* LPJZ-658, non-alcoholic steatohepatitis, bile acids, gut microbiota

## Abstract

Non-alcoholic steatohepatitis (NASH) is one of the most prevalent diseases worldwide; it is characterized by hepatic lipid accumulation, inflammation, and progressive fibrosis. Here, a Western diet combined with low-dose weekly carbon tetrachloride was fed to C57BL/6J mice for 12 weeks to build a NASH model to investigate the attenuating effects and possible mechanisms of *Lactiplantibacillus plantarum* LPJZ-658. Hepatic pathology, lipid profiles, and gene expression were assessed. The metabolomic profiling of the serum was performed. The composition structure of gut microbiota was profiled using 16s rRNA sequencing. The results show that LPJZ-658 treatment significantly attenuated liver injury, steatosis, fibrosis, and inflammation in NASH mice. Metabolic pathway analysis revealed that several pathways, such as purine metabolism, glycerophospholipid metabolism, linoleic acid metabolism, and primary bile acid biosynthesis, were associated with NASH. Notably, we found that treatment with LPJZ-658 regulated the levels of bile acids (BAs) in the serum. Moreover, LPJZ-658 restored NASH-induced gut microbiota dysbiosis. The correlation analysis deduced obvious interactions between BAs and gut microbiota. The current study indicates that LPJZ-658 supplementation protects against NASH progression, which is accompanied by alternating BA metabolic and modulating gut microbiota.

## 1. Introduction

Non-alcoholic fatty liver disease (NAFLD), which has an incidence of approximately 25% worldwide, has recently emerged as the most prevalent chronic liver disease [1]. Additionally, it is now the most rapidly increasing cause of liver-related mortality globally [2]. NAFLD refers to a wide range of diseases, from steatosis to non-alcoholic steatohepatitis (NASH), which are associated with excess lipid accumulation in the liver [3]. NASH is a crucial link in the progression of NAFLD, which combines steatosis, hepatocyte damage, and inflammation infiltration. It raises the risk of scarring or fibrosis, which ultimately leads to cirrhosis and hepatocellular carcinoma [4,5]. The pathogenesis of NASH is multifactorial and not fully understood. Previous studies have proposed that multiple factors, including lipotoxicity, inflammation, and bile acids (BAs) toxicity, are associated with NASH progression [6].

BAs are a group of steroid acids that play a vital role in nutrient absorption and act as key regulators of metabolism and immune homeostasis [7]. Primary BAs (cholic acid [CA], chenodeoxycholic acid [CDCA], and hyocholic acid [HCA]) are synthesized in the liver from cholesterol, stored in the gallbladder as the main component of bile, and secreted through the bile ducts into the intestine. There, the gut flora changes the primary BAs into secondary BAs (deoxycholic acid [DCA], lithocholic acid [LCA], and hyodeoxycholic acid [HDCA]). It is possible to measure each BA species either free or coupled to taurine or glycine [8]. In addition to being involved in lipid digestion and absorption, BAs also function as signaling molecules that participate in a variety of endocrine and metabolic processing, such as energy expenditure, lipid metabolism, glucose homeostasis, and inflammation [9,10]. BAs have been identified as essential signaling molecules in the pathogenesis of NASH, capable of fine-tuned gut–liver communication from the liver, and have been a therapeutic target for NAFLD prevention and/or treatment [11,12]. Globally, dysregulation in total BA levels and composition has been observed in both rodents and humans with NASH [13,14,15]. The metabolism of BAs is regulated by a broad spectrum of intestinal bacteria, which are involved in the deconjugation, dehydrogenation, and dehydroxylation of primary BAs [16]. Numerous studies have demonstrated that gut microbiota can influence host physiology by altering BAs’ metabolism [17].

Probiotics are living bacteria that provide health benefits by enhancing or repairing a healthy intestinal microbiome, which is widely used to prevent or treat gastrointestinal diseases. In recent years, several studies involving humans and animals have documented the benefits of probiotic supplementation with NAFLD, such as improving hepatic steatosis, oxidative damage, and inflammatory markers [18,19,20]. *Lactobacillus johnsonii* BS15 protected high fat diet (HFD) mice from hepatic steatosis and apoptosis by improving lipid peroxidation, supporting the antioxidant defense system, and correcting mitochondrial abnormalities [21]. The administration of VSL#3, a mixture of three genera of bacteria, limits inflammatory injury in the liver of NAFLD rats by inhibiting the production of tumor necrosis factor-α (TNF-α), inducible nitric oxide synthase (iNOS), and cyclooxygenase 2 (COX-2) [18]. In our previous studies, *Lactobacillus rhamnosus* GG (LGG) alleviated NAFLD by regulating the intestinal barrier function and improving lipid metabolism [22,23,24,25]. However, most studies on the role of probiotics have focused on the early stage of NAFLD, including liver steatosis and inflammation, while few studies have investigated the late stage, including liver steatohepatitis and fibrosis.

As a newly discovered strain, we have demonstrated that *Lactiplantibacillus plantarum* LPJZ-658 (CGMCC No. 22908) is safe with good probiotic properties. Notably, we found that LPJZ-658 contains a gene-encoding bile salt hydrolase (BSH), indicating that it may have the ability to regulate bile acid metabolism [26]. We further documented the effect of LPJZ-658 on improving the production performance and modulating the metabolism of both laying hens and broilers [27]. In the current study, we found that both LGG and LPJZ-658 were effective in the prevention of NASH, but LPJZ-658 was superior to LGG in liver injury and lipid metabolism. Therefore, utilizing serum metabolomics and gut microbiota analysis, this study aims to investigate the underlying mechanisms of LPJZ-658 in improving NASH. The results suggest that LPJZ-658 can alleviate the development of NASH by correcting BAs metabolism dysbiosis and modulating gut microbiota.

## 2. Results

### 2.1. LPJZ-658 Attenuates Hepatic Injury and Serum Lipid Metabolic Disorders in NASH Mice

To investigate the effects of LPJZ-658 on hepatic damage, we employed a mouse model for NASH, which resulted in severe hepatic steatosis, inflammation, and fibrosis. Macroscopically, livers showed hepatomegaly in response to NASH and were attenuated by LPJZ-658 (Figure 1A). Mice in the NASH group showed a significant rise in liver-to-body weight (LW/BW) and eWAT-to-body weight ratios (EW/BW) compared to the CON group. However, only LW/BW loss was affected by LPJZ-658 (Figure 1B). Liver injury was also reflected by increased serum ALT and AST levels in NASH mice compared to CON mice; moreover, this significantly decreased in NASH + LPJZ-658 mice (Figure 1C). The metabolic parameters of mice are shown in Figure 1D, where NASH caused a remarkable elevation in the serum levels of triglyceride (TG), total cholesterol (T-CHO), low-density lipoprotein cholesterol (LDL), and glycerol (GLY) alongside the demotion of high-density lipoprotein cholesterol (HDL). Although LPJZ-658 supplementation did not ameliorate serum HDL levels, it significantly decreased serum TG, T-CHO, LDL, GLY, and non-esterified fatty acid (NEFA) levels. These results demonstrate that LPJZ-658 attenuates NASH-related liver injury and dyslipidemia features.

### 2.2. LPJZ-658 Prevents the Development of Hepatic Steatosis in NASH Mice

The representative photomicrographs of H&E-stained liver slices of mice for all groups are represented in Figure 2A. The normal hepatic architecture was observed in the CON group mice, while the NASH group exhibited vacuoles relating to TG accumulation (macrogoticular steatosis and microvesicular steatosis), lobular inflammation (arrowhead), and ballooning degeneration (Mallory hyaline) of hepatocytes, unlike CON mice (Figure 2A). LPJZ-658-treated NASH mice reduced lipid accumulation with less ballooning degeneration and a considerably lower NAFLD activity score (NAS) than those of NASH mice (Figure 2B). Furthermore, compared to the CON group, bigger fat droplets in Oil Red O-stained liver slices of mice in the NASH group revealed that hepatic fat accumulation progressed markedly. The NASH + LPJZ-658 group exhibited lower lipid accumulation compared with the NASH group, suggesting that LPJZ-658 reduced lipid accumulation in the livers of NASH mice (Figure 2A). Consistent with this, hepatic TG and T-CHO levels were significantly increased in NASH mice, and LPJZ-658 treatment strikingly decreased hepatic TG levels (Figure 2C). Moreover, hepatic fatty acid synthesis-related genes (Fatty Acid Synthase [FAS], Stearoyl-CoA desaturase 1 [SCD1], cluster of differentiation 36 [CD36], sirtuins 1 [Sirt1], sterol-regulatory element-binding protein 1c [SREBP1c], and carbohydrate-responsive element-binding protein [ChREBP]), mRNA, and/or protein levels were significantly increased in NASH mice compared to CON mice, which strikingly decreased with LPJZ-658 treatment in NASH + LPJZ-658 mice. Additionally, the hepatic mRNA or protein levels of β-oxidation genes (peroxisome proliferator-activated receptor α [PPARα], proliferator-activated receptor gamma coactivator-1α [PGC-1α], and carnitine palmitoyltransferase 1 [CPT1]) were unaffected in NASH mice compared to CON mice; however, mRNA levels of PGC-1α and CPT1 notably increased in NASH + LPJZ-658 mice (Figure 2D–F).

### 2.3. LPJZ-658 Attenuates Hepatic Fibrosis and Inflammation in NASH Mice

Liver inflammation and fibrosis exacerbate NASH progression [28]. Representative pictures of Sirius Red staining from the mice of each group are shown in Figure 3A. LPJZ-658 prevented the development of hepatic fibrosis in NASH mice with severe fibrotic damage and noticeably increased collagen distribution regions (Figure 3B). Activated hepatic stellate cells (HSCs) constitute a significant generator of the extracellular matrix (ECM) in parenchymal liver disease, which is evidenced by increased alpha-smooth muscle actin (α-SMA). Next, we performed RT-PCR and immunoblotting analyses of α-SMA (Figure 3C,D). NASH-induced mice liver α-SMA mRNA and protein levels were significantly increased, whereas LPJZ-658 markedly decreased both mRNA and protein levels. To determine how LPJZ-658 prevented hepatic fibrosis in NASH mice, we examined the expression of transforming growth factor-β (TGF-β), a master regulator of fibrogenesis, and targeted genes such as collagen 1 alpha 1 (Col1a1) in the liver (Figure 3C,D). NASH treatment significantly upregulated the hepatic mRNA and protein levels of TGF-β compared to the CON mouse. However, increases in hepatic TGF-β were greatly decreased by the administration of LPJZ-658. In addition, NASH treatment increased hepatic Col1a1 mRNA levels, which were reduced by LPJZ-658 supplementation. Inflammation in the liver tissue has been described as a driver for the development of hepatic fibrosis. We next assessed the effect of LPJZ-658 on inflammation in NASH mice. Our results indicate that the serum levels of interleukin 6 (IL-6), TNFα, and monocyte chemotactic protein 1 (MCP1) were increased in NASH mice, while LPJZ-658 supplementation significantly reversed all of these phenomena (Figure 3E). Furthermore, the NASH group exhibited significant elevations in the mRNA levels of IL-6, interleukin-1β (IL-1β), and TNFα compared to the CON group. On the other hand, NASH + LPJZ-658 groups significantly reversed HASH-induced increases in these liver mRNA expressions (Figure 3F).

### 2.4. Serum Metabolome Profile and Biomarker Annotation

Serum extracts were analyzed by UHPLC-Q-Orbitrap/MS, and the base peak chromatograms (BPC) of different groups are shown in Appendix A. The peaks differed considerably in retention time and peak intensity. However, since each chromatogram contained many ions, the identification of metabolites required multivariate statistical analysis. After MS-Dial data processing, a dataset containing sample information, RT *m*/*z*, and peak intensities was created for statistical analysis, and an OPLS-DA model was constructed to visualize the classification patterns. As shown in Figure 4A,B, score plots were created from data obtained by LC-MS in both positive and negative ion modes from serum extracts. We could observe that the CON group was clearly distinguished from the NASH group. After treatment with LPJZ-658, the points in the NASH + LPJZ-658 group in the negative ion mode showed complete separation from the NASH group. There was still a partial crossover in the positive ion mode, suggesting that treatment with LPJZ-658 could partially restore metabolic changes in the NASH group.

To obtain the serum metabolites that contributed most to sample separation in both CON and NASH groups, as shown in Figure 4C–H, we selected feature markers with VIP > 1 for screening. The biomarkers were then further highlighted with fold change (FC > 1.2 or FC < 0.83) and a one-way ANOVA *p*-value (*p* < 0.05). The identification of these biomarkers was achieved through database searches using accurate mass spectrometry and tandem mass spectrometry information.

Using the ion at RT 9.20 min m/z 391.2845 as an example, the MS spectrum in Figure 4I showed that the ion at m/z 391.2845 belonged to [M-H]^−^. As shown in Figure 4I–K, this compound produced four fragment ions, including m/z 373.2722 [M-H-H_2_O]^−^, 345.2785 [M-H-HCOOH]^−^, 327.2721 [M-H-HCOOH-H_2_O]^−^. The compound was identified as deoxycholic acid (DCA) by searching the HMDB database. A total of 178 ions were identified and listed in Appendix A using the above procedure.

### 2.5. Metabolic Pathway Analysis

MetaboAnalyst is a free web-based portal for performing statistical analysis, biomarker analysis, pathway analysis, etc. As shown in Appendix A, 178 biomarkers were identified in both the CON and NASH groups. Among the 178 differential metabolites, 55 were upregulated metabolites, and 123 were downregulated metabolites. These differential metabolites were plotted with a heatmap to provide a more comprehensive picture of the biochemical differences between the two groups. As shown in Figure 4L, each column represents a sample, and each line represents a metabolite with its HMDB number. The details of these metabolites are listed in Appendix A. The identified biomarkers were imported into MetaboAnalyst to conduct pathway analysis. The KEGG database for the annotation of potential biomarkers and identification of their associated metabolic pathways, as well as the potential relationships between these biomarkers, is presented in the form of metabolic network pathways. As shown in Figure 4M, the results indicate that these biomarkers affect multiple metabolic pathways, including purine metabolism, glycerophospholipid metabolism, linoleic acid metabolism, primary bile acid biosynthesis, etc. It indicated that LPJZ-658 might alleviate the metabolic changes caused by NASH by adjusting these pathways.

### 2.6. LPJZ-658 Altered the BAs Metabolism Profiles in NASH Mice Serum

Primary BAs, including CDCA and CA in humans and rodents and CDCA and HCA in pigs, are generated in the liver and subsequently conjugated to taurine or glycine before being excreted into the intestinal tract after the ingestion of food. Once in the ileum, intestinal bacteria covert primary BAs into more toxic non-conjugated secondary BAs, including numerous chemically different secondary BAs such as DCA, LCA, HDCA, and their conjugated forms and isomers [29]. The results of pathway analysis show that bile acid metabolism plays an important role in regulation (Figure 4M). Therefore, we focused on changes in bile acids in four groups of the serum. The intensity of BAs is shown in Appendix A. Primary BAs (CA and HCA), secondary BAs (DCA, LCA, and HDCA), and conjugated BAs (glycocholic acid [GCA], taurocholate acid [TCA], taurohyocholic acid [THCA], and taurohyodeoxycholic acid [THDCA]) were significantly higher in the NASH group compared to the CON group. After treatment with LPJZ-658, bile acid-related compounds were all reduced to varying degrees, with a tendency to converge to the CON group. In particular, LPJZ-658 supplementation significantly reduced the contents of LCA, GCA, and THDCA. These results suggest that LPJZ-658 has the potential to alleviate WD/CCl4-induced NASH by regulating bile acid metabolism.

### 2.7. Effect of LPJZ-658 Gut Microbiota in NASH Mice

Alterations in BA metabolism profiles are closely related to gut bacteria changes. Thus, fecal microbiota compositions were examined by 16S rRNA gene sequencing to assess the effects of LPJZ-658 supplementation on the gut bacteria of NASH mice. As shown in Figure 5A, the rarefaction curve of each feces sample tended to be flat, suggesting a credible sequencing result.

The PCoA results shown in Figure 5B, representing beta diversity, reflect how gut microbiota compositions changed in the NASH and NASH + LPJZ-658 groups. Linear discriminant effect size (LEfSe) analysis was performed to distinguish the significant features of gut microbiota. The results (Figure 5C) showed that the LPJZ-658 group had the highest abundance of Bacteroidota. The gut microbiota in the NASH group dramatically increased in Verrucomicrobiota at the phylum level and Akkermansia at the genus level, while NASH + LPJZ-658 elevated abundances of Lactobacillus and Faecalibaculum at the genus level.

Based on the classification and analysis of reads, gut bacteria compositions at the phylum and genus levels are shown in Figure 5D and Figure 5E, respectively. At the phylum level, the results showed that Firmicutes and Bacteroidota were the dominant phyla in all groups, representing more than 60% of relative bacterial abundance. This increased relative abundances of Verrucomicrobia in NASH (22.51%) and NASH + LPJZ-658 (18.37%) groups (Figure 5D). In addition, the relative abundance of the main microbiota at the genus level is shown in Figure 5E,F. In contrast to the NASH group, relative abundances of Oscillibacter and Mucispirillum were significantly decreased, and Faecalibaculum was observed with a significant increase in the NASH + LPJZ-658 group.

### 2.8. Correlations between BAs Metabolism Profiles and Gut Bacteria

To investigate the relationship between BA metabolism profiles and gut microbiota, we correlated serum BA levels with gut microbiota at the genus and OTU levels. As shown in Figure 6, Oscillibacter was positively correlated with CDCA, GCA, DCA, HDCA, CA, LCA, TCA, and THDCA, while Mucispirillum was positively correlated with CA, DCA, LCA, HDCA, GCA, GDCA, TCA, THDCA, and CDCA.

## 3. Discussion

NASH is caused by a persistent state of lipotoxicity in which toxic lipid species accumulate and induce hepatocyte cell death and the activation of sterile innate immunological pathways, resulting in a vicious cycle of inflammation, liver damage, and, eventually, fibrosis. In recent years, growing evidence has established the therapeutic function of probiotics in the prevention/treatment of NAFLD [30,31]. In previous studies, we showed that LPJZ-658 supplementation improved the productivity of laying hens and broilers by modulating lipid metabolism and gut microbiota [27]. In the present study, we evaluated the impact of LPJZ-658 on NASH pathology. This strain was first reported to improve NAFLD; therefore, we selected the widely used probiotic strain Lactobacillus rhamnosus GG (LGG) as a potential probiotic [32], which has previously been demonstrated to reduce hepatic lipid accumulation and liver injury by regulating intestinal barrier function and improving lipid metabolism in previous studies [22,23,24]. Moreover, LGG supplementation prevents liver fibrosis by suppressing BA denovo synthesis and enhancing BA excretion, which results in less BA-induced liver injury and fibrosis in mice [33]. Here, we targeted NASH-induced liver injury and lipid metabolism disorder by the administration of either LGG or LPJZ-658. After the implementation of LGG or LPJZ-658, we found that the liver/body weight ratio, serum TG, T-CHO, LDL levels, and hepatic collagen area were significantly lower than the NASH group (Appendix A). However, LPJZ-658, but not LGG, decreased serum ALT, AST, GLY, and NEFA levels compared to the NASH group (Appendix A). Moreover, LGG and LPJZ-658 led to a decline in liver NAS and TG content (Appendix A). It is noteworthy that, by comparison, the liver NAS and TG content were lower in NASH + LPJZ-658 mice compared to NASH + LGG mice. These results mean that both LGG and LPJZ-658 can alleviate NASH, and LPJZ-658 is superior to LGG in some parameters.

In this investigation, NASH was successfully induced in mice, as demonstrated by aberrant liver function tests (Figure 1C), lipid profiles (Figure 1D), and hepatic histological characteristics. NASH features include macro- and micro-vesicular steatosis, varying degrees of inflammation, and increasing fibrosis (Figure 2A–C). The pathogenesis of NAFLD and its progression is complicated, with some unanswered questions. Classical theories of the pathogenesis of NAFLD and NASH have been described in terms of the “two hit hypothesis” [34]. Excess hepatic lipid accumulation is considered the “first hit”, which causes increased liver weight [35]. In the present study, LPJZ-658 reduced hepatic lipid content in the livers of NASH mice (Figure 2C). Hepatic lipid content is determined by the balance of lipid input and output [36,37]. Hepatic lipid input is mainly derived from dietary fat intake, the lipolysis of adipose tissue, and de novo lipogenesis [38,39,40]. In addition, the β-oxidation of fatty acids plays a central role in hepatic lipid output [41]. Moreover, LPJZ-658 inhibited fatty acid uptake and synthesis-related genes [42,43] FAS, SCD1, CD36, SREBP1c, ChREBP mRNA expression promoted lipid β-oxidation genes and the related genes’ PGC-1α and CPT1 mRNA expression, suggesting that LPJZ-658 may decreased hepatic accumulation and steatosis by inhibiting the de novo synthesis and uptake of fatty acids while accelerating fatty acid β-oxidation.

A fatty liver is more susceptible to “second hit”; it promotes hepatic injury, inflammation, and fibrosis. A characteristic feature of NASH is the presence of hepatic inflammation, and persistent inflammation in the liver is thought to drive fibrosis development [44]. Several studies have provided evidence that probiotics have anti-inflammatory effects that can contribute to their clinical benefits in NAFLD [45,46]. IL-6, TNF-α, MCP1, and IL-1β are the inflammatory cytokines responsible for NASH progression, which play an important role in hepatic fibrosis [47,48,49,50]. In the current study, and compared with the NASH group, LPJZ-658 attenuated the secretion of IL-6, TNF-α, and MCP1, along with inhibiting liver IL-6, TNF-α, and IL-1β mRNA expression (Figure 3E,F). These results suggest that NASH leads to the development of inflammation, which is alleviated by LPJZ-658 treatment. HSCs are primarily responsible for the advancement of hepatic fibrosis following hepatocyte injury and inflammation by altering their features from a quiescent to an active state [51]. HSC activation leads to an increase in ECM synthesis and the formation of a fibrous scar [52]. In the liver, α-SMA expression is considered a reliable marker of hepatic HSCs, and TGF-β signaling participates in fibrogenic responses through HSC activation [53]. Our results observed a significant increase in α-SMA and TGF-β signals in the liver of NASH mice, both of which were reduced by LPJZ-658 treatment (Figure 3C,D). Taken together, we interpreted these findings as supporting the idea that LPJZ-658 is involved in improvements of hepatic inflammation and fibrosis.

NAFLD is a complex condition that affects many systems of the body. Many pathways, such as BAs and fibroblast growth factors, have emerged as significant pathophysiological players in NASH [54,55,56]. In our serum metabolomics study, LPJZ-658 affects multiple metabolic pathways, especially bile acid biosynthesis, which has attracted our attention. BAs are a major component of bile, which are generated by hepatocytes using cholesterol and excreted into the proximal small intestine to help solubilize and absorb lipophilic nutrients and vitamins following food intake [57,58]. According to studies in NAFLD patients, overall BA levels were elevated, primarily by conjugated and secondary BAs [59,60]. Moreover, primary bile acids, including TCA and GCA, were enriched in NASH mice, which is consistent with high-plasma TCA and GCA previously reported in human NASH patients compared with healthy subjects [60], linking the gut and liver to impact hepatic lipid accumulation and inflammation [61]. BA accumulation induces parenchymal liver injury and contributes to the progression of NASH [62]. Currently, obeticholic acid (OCA) is the only FDA-approved drug for NASH, as it regulates the expression of transcription factors that reduce bile acid synthesis and hepatic steatosis [63]. Here, we analyzed the serum parameters of BA metabolism. These results clearly illustrate that LPJZ-658 effectively improves NASH-induced BA metabolism abnormalities (Appendix A). In agreement with our observation, a previous study reported that the administration of probiotics reverses NASH in mice due to dysregulated BA synthesis and dysbiosis [64].

The gut microbiota is closely linked to the development of NAFLD and is essential for intestinal BA deconjugation and excretion [65]. Therefore, the interaction between BAs and gut microbiota is also involved in the pathogenesis of NAFLD. In this process, gut bacteria-derived BSH plays a key role. Bacteria with high BSH activity promote BA deconjugation [66]. In our experiment, we analyzed the gut bacteria phylum and genus levels. Gut microbiota analysis showed that phylum levels of Firmicutes and Actinobacteria, known as phyla that harbor bacteria with high BSH activity [67], were significantly increased in NASH treatment and altered by LPJZ-658 treatment. LPJZ-658 treatment altered gut microbiota with an enrichment of the gut flora with BSH-containing phyla. We analyzed the LPJZ-658 genome and found that LPJZ-658 contained a gene encoding BSH [26]. Therefore, LPJZ-658 itself may have high BSH activity. In addition, at the genus level, the abundance of Oscillibacter and Mucispirillum was increased in the NASH group, and LPJZ-658 treatment reduced the relative abundances of these intestinal florae. Recent findings have reported that Oscillibacter and Mucispirillum are positively related to high-fat diet-induced obesity and NAFLD [68,69]. Furthermore, the present work suggested that Oscillibacter may be crucial in HFD-induced intestinal dysfunction [70]. Moreover, the enrichment of Mucispirillum is positively correlated with TUDCA, TCDCA, TCA, and GCA in cholesterol-induced NAFLD-HCC mice [50]. Thus, our study revealed that LPJZ-658 could regulate BA metabolism by improving gut microbiota in NASH mice.

## 4. Materials and Methods

### 4.1. Probiotic Strain

*Lactiplantibacillus plantarum* LPJZ-658 (NCBI no. SRR22306760) was isolated from fermented dairy products containing probiotics and deposited in the China General Microbiological Culture Collection Center (CGMCC No. 22908, Beijing, China). The freeze-dried powder of LPJZ-658 was provided by Tianshu Yaoyuan (Tianjin) Biotechnology Co., Ltd. (Tianjin, China). Before the daily oral supplement, LPJZ-658 powder was suspended in sterile physiological saline (PBS) and adjusted to 5 × 10^9^ CFU/mL.

### 4.2. Animals and Treatments

Eight- to ten-week-old male C57BL/6J mice were purchased from Jilin Genet-Med Biotechnology Co., Ltd. (Jilin, China) and maintained in a 12 h light/dark cycle environment. All mice were randomly divided into four groups after 1 week of acclimation: a control (CON) group, LPJZ-658 group, NASH group, and NASH + LPJZ-658 group for 12 weeks. The CON group and LPJZ-658 group were fed a normal chow diet (ND) (AIN-93G, Xiao Shu You Tai [Beijing] Biotechnology Co., Ltd., Beijing, China) and normal tap sterilized water. The NASH group and NASH + LPJZ-658 group were fed a Western Diet (WD) [71] (Xiaoshuyoutai [Beijing] Biotechnology Co., Ltd., Beijing, China) and a high sugar solution (18.9 g/L glucose and 23.1 g/L fructose [Sigma-Aldrich, Louis, MO, USA]). The CON group and LPJZ-658 group were injected intra-peritoneally (I.P.) with corn oil, whereas the NASH group and NASH + LPJZ-658 group were injected I.P. with carbon tetrachloride (CCl4, Shanghai Titan Scientific Co., Ltd., Shanghai, China) at a dose of 0.32 μg/g of body weight once/week, respectively. In addition, the LPJZ-658 and NASH + LPJZ-658 groups were administered oral gavage daily with 1 × 10^9^ CFU/day per mouse in the last 4 weeks of the experiment.

After 12 weeks of treatment, the mice were euthanized under anesthesia using Avertin (2, 2, 2-tribromoethanol, Sigma-Aldrich, Louis, MO, USA). Mice cecal contents were collected and stored at −80 °C and then transported to the laboratory for gut microbiota analysis. Liver samples were collected and processed for histological, lipid profiles, and gene expression analysis. Serum samples were collected after removing blood cells and were processed for serological and metabolomics analysis (Figure 7). For the animal study, *n* = 6–12 per group.

### 4.3. Statistical Analysis and Biomarker Screening

Pathway enrichment analysis was performed by MetaboAnalyst 5.0 service. Data filtering using the interquartile range (IQR), normalization by sum pairs for normalization, and pareto scaling for data scaling were followed by multi-factor analysis. Principal component analysis (PCA) and orthogonal partial least squares discriminant analysis (OPLS-DA) were then used to identify differentiated metabolites between the two groups.

Differences in quantitative data, expressed as the mean ± SEM with statistical significance denoted by * *p* < 0.05, ** *p* < 0.01, *** *p* < 0.001 between groups were compared by two-way ANOVA with Bonferroni’s post hoc test using GraphPad Prism 7 (GraphPad Software Inc., La Jolla, CA, USA).

Additional methods are described in the Appendix A.

## 5. Conclusions

In summary, our present study demonstrates that LPJZ-658 alleviated hepatic fat accumulation, inflammation, and fibrosis in a Western diet combined with the CCl4-induced NASH mouse model. Briefly, LPJZ-658 treatment regulated BA metabolism and changed the gut microbiota composition, which likely contributed to hepatic steatosis, inflammation, and fibrosis prevention. To the best of our knowledge, this is the first report on the protective effects of LPJZ-658 against the development of NAFLD associated with BA metabolism.

## Figures and Tables

**Figure 1 ijms-24-13997-f001:**
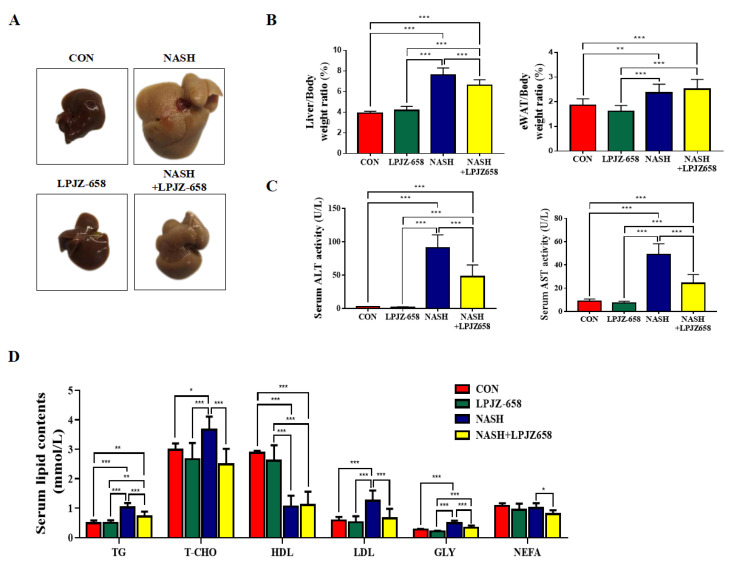
LPJZ-658 alleviates hepatic injury and serum lipid homeostasis in mice treated with western diet/carbon tetrachloride (WD/CCl4). (**A**) Representative photographs of mouse livers. (**B**) Liver-to-body weight ratio (**left**), eWAT-to-body weight ratio (**right**) of mice. (**C**) Serum levels of ALT and AST. (**D**) Serum TG, T-CHO, HDL, LDL, GLY, and NEFA. Data presented indicate the mean ± SEM (* *p* < 0.05, ** *p* < 0.01, and *** *p* < 0.001).

**Figure 2 ijms-24-13997-f002:**
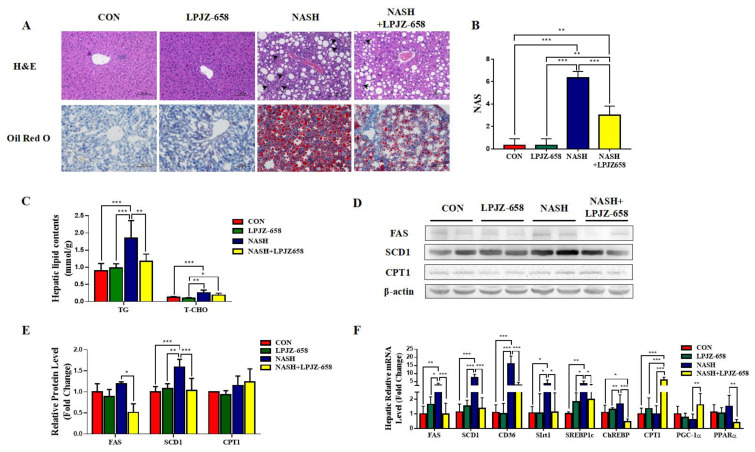
LPJZ-658 represses hepatic steatosis in mice treated with WD/CCl4. (**A**) Representative images of H&E and Oil Red O staining of liver sections from the indicated mice. Scale bar, 100 µm. (**B**) NAFLD activity score (NAS). (**C**) Lipid (TG and T-CHO) levels in the hepatic. Immunoblots (**D**) and quantification (**E**) of hepatic FAS, SCD1, and CPT1. β-actin served as a loading control. (**F**) Quantitative PCR analyses of hepatic mRNA levels in genes related to fatty acid metabolism (FAS, SCD1, CD36, Sirt1, SREBP1c, ChREBP, CPT1, PGC-1α, and PPARα). Data presented indicate the mean ± SEM (* *p* < 0.05, ** *p* < 0.01, and *** *p* < 0.001).

**Figure 3 ijms-24-13997-f003:**
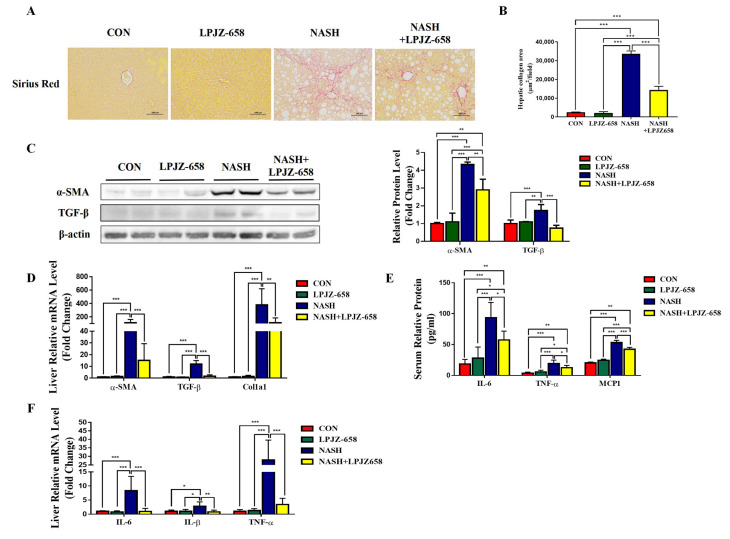
Attenuation effect of LPJZ-658 on fibrosis and inflammation in mice treated with WD/CCl4. Representative images of Sirius Red staining of liver sections (**A**) from the indicated mice and the quantitation of Sirius Red staining were calculated (**B**). Scale bar, 100 µm. (**C**) Immunoblots (**left**) and quantification (**right**) of hepatic α-SMA and TGF-β. β-actin served as a loading control. (**D**) Quantitative PCR analyses of hepatic mRNA levels of genes related to fibrosis indicators (α-SMA, TGF-β, and Col1a1). (**E**) Serum levels of IL-6, TNF-α, and MCP1. (**F**) Quantitative PCR analyses of hepatic mRNA levels in genes related to inflammation (IL-6, IL-1β, and TNF-α). Data presented indicate the mean ± SEM (* *p* < 0.05, ** *p* < 0.01, and *** *p* < 0.001).

**Figure 4 ijms-24-13997-f004:**
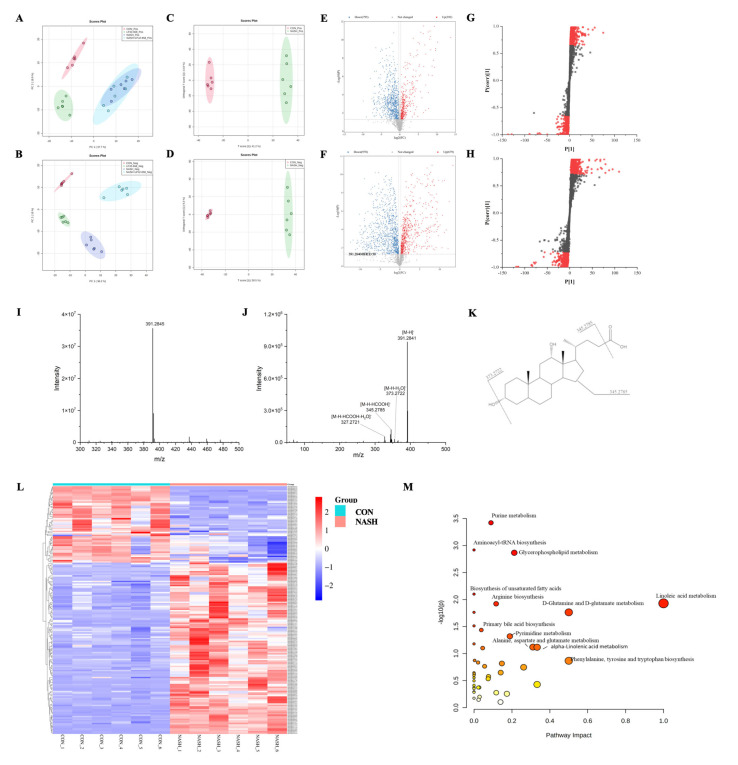
Effects of LPJZ-658 on serum metabonomic profiling by LC-MS. PCA score plots of the CON group, LPJZ-658 group, NASH group and NASH + LPJZ-658 group based on the data acquired from 50% methanol in water extracts ((**A**) ESI+ and (**B**) ESI−). OPLS-DA score plots/Volcano plots/S-plot of CON group and NASH group based on the data acquired from 50% methanol in water extracts ((**C**,**E**,**G**), ESI+, and (**D**,**F**,**H**), ESI−). Mass spectra of the deoxycholic acid (DCA) at RT 9.20 min *m*/*z* 391.2840, full scan mass spectrum (**I**), tandem mass spectrum (**J**), and structural formula (**K**). (**L**) Heatmap and hierarchical cluster analysis of differential metabolites between the CON group and NASH group. (**M**) The pathway analysis visualized by bubbles plot.

**Figure 5 ijms-24-13997-f005:**
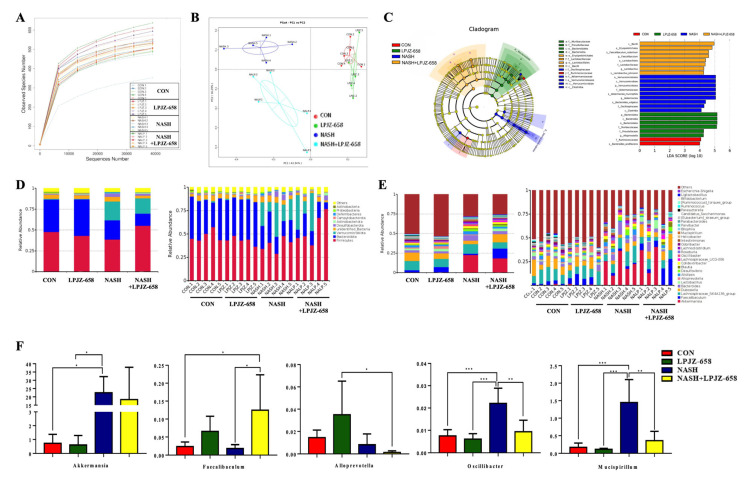
Effects of LPJZ-658 treatment on gut microbiota. (**A**) Rarefaction curves of feces samples; (**B**) Principal coordinates analysis (PCoA) of gut microbiota; (**C**) Results of LEfSe analysis (scores > 4 and significance of *p* < 0.05 as determined by Wilcoxon’s signed–rank test) showing bacterial taxa with differentially abundance among the groups studied. (**D**) Relative abundance of microbiota at the phylum level. (**E**) Relative abundance of microbiota at the genus level. (**F**) Relative abundance of Akkermansia, Faecalibaculum, Alloprevotella, Oscillibacter, and Mucispirillum. Data presented indicate the mean ± SEM (* *p* < 0.05, ** *p* < 0.01, and *** *p* < 0.001).

**Figure 6 ijms-24-13997-f006:**
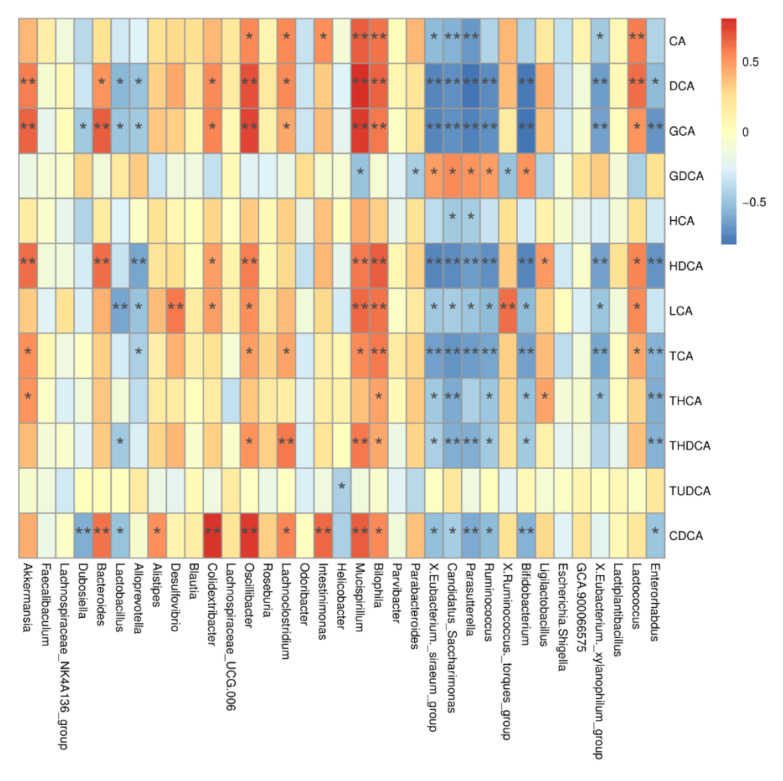
Correlation between differential BAs and differential gut microbiota. CA, cholic acid; DCA, deoxycholic acid; GCA, glycocholic acid; GDCA, glycodeoxycholic acid; HCA, hyocholic acid; HDCA, hyodeoxycholic acid; LCA, lithocholic acid; TCA, taurocholic acid; THCA, taurohyocholic acid; THDCA, taurohyodeoxycholic acid; TUDCA, tauroursodeoxycholic acid; CDCA, chenodeoxycholic acid. Data presented indicate the mean ± SEM (* *p* < 0.05 and ** *p* < 0.01).

**Figure 7 ijms-24-13997-f007:**
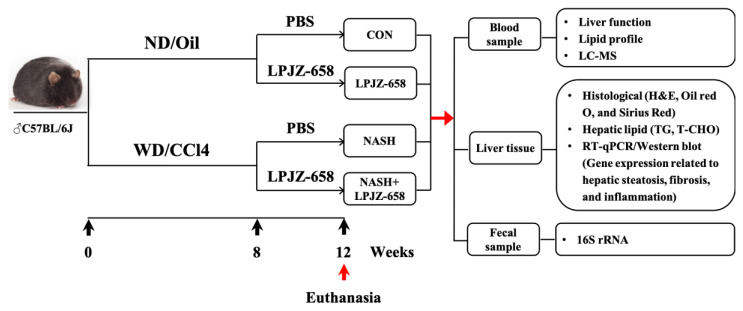
Schematic diagram of the experimental procedure used to examine the protective role of LPJZ-658 in mice treated with normal chow diet/corn oil (ND/Oil) or WD/CCl4 for 12 weeks. Mice were intragastrically administered PBS or LPJZ-658 (1 × 10^9^ CFU) once daily at the beginning of week 8 for 4 weeks.

## Data Availability

The datasets used and/or analyzed during the current study are available from the corresponding author on reasonable request.

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
