# Peer review of "Lactiplantibacillus plantarum LPJZ-658 Improves Non-Alcoholic Steatohepatitis by Modulating Bile Acid Metabolism and Gut Microbiota in Mice"

_ijms, 2023, doi:10.3390/ijms241813997_

Round 1

Reviewer 1 Report

 Comments to the Authors of manuscript ID: ijms-2586433 entitled “Lactiplantibacillus plantarum LPJZ-658 Improves Non-Alcoholic Steatohepatitis by Modulating Bile Acid Metabolism and Gut Microbiota in Mice”.

The authors have not presented the methods employed in their experiment according to established scientific standards. Moreover, in the results section, they discuss outcomes that lack adequate support from the obtained results and have not been elucidated in the materials and methods section, such as the observation of liver pathological changes and collagen content. A significant revision of the manuscript is warranted. The use of metabolomics and gene-level analysis, though mentioned, does not mitigate this concern. Additionally, the methods utilized lack a comprehensive description, and even those provided in the supplementary material are insufficiently elucidated. The figures lack arrows to illustrate the described changes. Notably, the PSR staining method is utilized to assess collagen content, albeit in a dark field setting.

1. L 23 – unclear

2. L 31 – unclear

3. L 44- pathogenesis of what?

4. L 47 – bile acids? It should be clarify

5. L 55- they are components of bile that is released to intestine

6. L 67 – if probiotics – are

7. L 73-74- if the two stages are mentioned, the entire process should be described shortly

8. L 77 – this part should be rephrased substantially. There should be presented basal hypothesis and the goal of the study and methods that were used to check the hypothesis

9. L 85  - called LPJZ-658

10. L 86 – it is Lactiplantibacillus plantarum

11. L 87 – unclear

12. L 89 – what is WD; what is CC14

13. Fig. 1 A should be moved to the material and methods

14. L 88- 90 – material and methods

15. L 379-381- unfinished sentence

16. part 4.1 – if in L 383 is described the way by which it is given, the dose should be provided with CFU

17. L 390 – small letter

18. L 389 – the reference for this Western Diet used needed

19. L 392- is it tetrachloromethane? It should be clarify

20. L 386- 399- chaotic description. It should be described in line with time

21. L 401 – feces from what part of GIT? And then what?

22. L 402 Liver and serum samples subjected to gut microbiota analysis?

23. n of animals in each group should be indicated

24. L 402 – what part of liver was subjected to histological analysis.

25. the description of any methodology is very poor and this way is not accepted

26. part of 2.2 the lack of proper histopathology of liver and Figure 2A is very poor and not adequately described.

27. L 135 – Sirus Red staining serves to described collagen content but not in light

28. Figure 3A does not show collagen

29. L 137 and fig. 3b – there is lack the description how collagen content was analyzed. The result should be proven

30. L 149- there is no inflammation

Author Response

We would like to express our gratitude for the opportunity to revise our manuscript entitled “Lactiplantibacillus plantarum LPJZ-658 Improves Non-Alcoholic Steatohepatitis by Modulating Bile Acid Metabolism and Gut Microbiota in Mice” (Manuscript ID: ijms-2586433). We sincerely thank all the reviewers for their valuable feedback, which we have carefully considered and used to improve the quality of our work.

We have addressed the reviewers’ comments in a point-by-point manner below, and we have made the necessary revisions to the manuscript. Specific concerns raised by the reviewers have been numbered for clarity. Our responses are given in normal font, and changes/additions to the manuscript are highlighted in red.

Thank you again for your time and consideration.

Response to the comments of Reviewer 1

Point 1: L 23 – unclear

Response 1: We feel sorry for our unclear description. We have made modifications of this sentence in the revised manuscript.

Point 2: L 31 – unclear

Response 2: We feel sorry for our unclear description. We have made modifications of this sentence in the revised manuscript.

Point 3: L 44- pathogenesis of what?

Response 3: Thanks for your correction. We have corrected this sentence.

Point 4: L 47 – bile acids? It should be clarify

Response 4: We feel sorry for our unclear description. We have clarified it.

Point 5: L 55- they are components of bile that is released to intestine

Response 5: Thank you for the question. Yes, bile acids (BAs) are amphipathic molecules that facilitate the absorption of dietary fat and lipophilic vitamins in the small intestine. BAs are synthesised in the liver as primary BAs (cholic acid [CA], chenodeoxycholic acid [CDCA], hyocholic acid [HCA]), secreted in the bile and the intestine where they are transformed by the gut flora into secondary BAs (deoxycholic acid [DCA], lithocholic acid [LCA], hyodeoxycholic acid [HDCA], ursodeoxycholic acid [UDCA]). Each BA species can be measured as free or conjugated to taurine or glycine. In humans, more than 90% of BAs are reabsorbed in the intestine and returned to the liver via the portal vein in an enterohepatic cycle. We have made a short description.

Point 6: L 67 – if probiotics – are

Response 6: Thanks for your correction. We have corrected this mistake.

Point 7: L 73-74- if the two stages are mentioned, the entire process should be described shortly

Response 7: Thank you for your suggestions. We have added brief description in the revised manuscript.

Point 8: L 77 – this part should be rephrased substantially. There should be presented basal hypothesis and the goal of the study and methods that were used to check the hypothesis

Response 8: We feel great thanks for your professional review work on our article. According to your comments, we have made modifications to our previous description in the revised manuscript.

Point 9: L 85  - called LPJZ-658

Response 9: Thanks for your correction. We have corrected it.

Point 10: L 86 – it is Lactiplantibacillus plantarum

Response 10: Thanks for your correction. We have corrected it.

Point 11: L 87 – unclear

Response 11: We feel sorry for our unclear description. We have made modifications of this sentence in the revised manuscript.

Point 12: L 89 – what is WD; what is CC14

Response 12: We have added the full name of WD and CCl4 in the revised manuscript.

Point 13: Fig. 1 A should be moved to the material and methods

Response 13: We sincerely appreciate the valuable comments. We have moved Fig. 1A to the material and methods.

Point 14: L 88- 90 – material and methods

Response 14: We sincerely appreciate the valuable comments. We have moved this part to the material and methods.

Point 15: L 379-381- unfinished sentence

Response 15: Thanks for your correction. We have corrected this sentence.

Point 16: part 4.1 – if in L 383 is described the way by which it is given, the dose should be provided with CFU

Response 16: Thanks for your valuable comments. We have corrected it.

Point 17: L 390 – small letter

Response 17: Thanks for your correction. We have corrected this mistake.

Point 18: L 389 – the reference for this Western Diet used needed

Response 18: Thanks for your suggestion. We have cited the references into the revised manuscript.

Point 19: L 392- is it tetrachloromethane? It should be clarify

Response 19: We are sorry for our limited description. Yes, the full name of CCl4 is carbon tetrachloride, also known as tetrachloromethane.

Point 20: L 386- 399- chaotic description. It should be described in line with time

Response 20: Thanks for your valuable comments. We have re-written “Materials and Methods” according to the Reviewer’s suggestion.

Point 21: L 401 – feces from what part of GIT? And then what?

Response 21: We are sorry for our limited description. Mice cecal contents were collected and stored at -80 ℃, then transported to the laboratory for gut microbiota analysis.

Point 22: L 402 Liver and serum samples subjected to gut microbiota analysis?

Response 22: We feel sorry for our carelessness. We have corrected this mistake in the revised manuscript.

Point 23: n of animals in each group should be indicated

Response 23: Thanks for your valuable comments. We have added the description in the revised manuscript.

Point 24: L 402 – what part of liver was subjected to histological analysis.

Response 24: In this study, we collected the same part of the left liver lobe from mice for histological analysis.

Point 25: the description of any methodology is very poor and this way is not accepted

Response 26: We feel great thanks for your professional review work on our article. We have re-written “Materials and Methods” according to the Reviewer’s suggestion.

Point 26: part of 2.2 the lack of proper histopathology of liver and Figure 2A is very poor and not adequately described.

Response 26: We feel sorry for the unclear histopathology results of Fig 2A. Therefore, we added black arrows to the H&E staining to indicate inflammatory cell infiltration. In addition, we have also described this part of the results in more detail.

Point 27: L 135 – Sirus Red staining serves to described collagen content but not in light

Response 27: Thanks for your valuable comments. In our study, Sirius Red stain were performed according to the manufacturer’s instructions (Servicebio, Wuhan, China) as described previously 1. Collagen fibers appeared red inunder normal light microscopy and other tissue components are stained yellow.

Point 28: Figure 3A does not show collagen

Response 28: Thanks for your valuable comments. Sirius Red is a strong acidic dye, which easily combines with the alkaline groups in the collagen molecule and adsorbs firmly. According to the manufacturer’s instructions (Servicebio, Wuhan, China), collagen fibers appeared red in Sirius Red staining 2.

Point 29: L 137 and fig. 3b – there is lack the description how collagen content was analyzed. The result should be proven

Response 29: We are sorry for our limited description. The percentage of fibrosis area in Sirius Red staining was measured by Image J 1.8.0.

Point 30: L 149- there is no inflammation

Response 30: It is possible that the picture of Fig 2A is too small to show clearly. In fact, H&E-stained sections revealed that the liver had inflammatory foci, and we added black arrows to indicate inflammatory cell infiltration.

Best Wishes to you!

Yours sincerely,

Cuiqing Zhao

References

  1. Liang QS, Xie JG, Yu C, et al. Splenectomy improves liver fibrosis via tumor necrosis factor superfamily 14 (LIGHT) through the JNK/TGF-beta1 signaling pathway. Exp Mol Med 2021;53:393-406.
  2. Peng C, Tu G, Yu L, et al. Murine Chronic Pancreatitis Model Induced by Partial Ligation of the Pancreatic Duct Encapsulates the Profile of Macrophage in Human Chronic Pancreatitis. Front Immunol 2022;13:840887.

Reviewer 2 Report

In the present experimental study Liu et al showed, in a murine model of steatohepatitis, that probiotic LPJZ-658 formulation restored NASH-related dysbiosis and improved serum lipid and inflammatory cytokines profile. This is an excellent, well-rounded study, and I have only few minor criticisms that should be addressed:

1) The most important one is that several experiments have not been described in the Methods section (in particular those reported in paragraph 2.1, 2.2 and 2.3). Therefore this part should be re-written.

2) Check for some exponential figure (e.g. 109 page 2 line 83).

3) Please explain NEFA acronym.

see above

Author Response

We would like to express our gratitude for the opportunity to revise our manuscript entitled “Lactiplantibacillus plantarum LPJZ-658 Improves Non-Alcoholic Steatohepatitis by Modulating Bile Acid Metabolism and Gut Microbiota in Mice” (Manuscript ID: ijms-2586433). We sincerely thank all the reviewers for their valuable feedback, which we have carefully considered and used to improve the quality of our work.

We have addressed the reviewers’ comments in a point-by-point manner below, and we have made the necessary revisions to the manuscript. Specific concerns raised by the reviewers have been numbered for clarity. Our responses are given in normal font, and changes/additions to the manuscript are highlighted in red.

Thank you again for your time and consideration.

Response to the comments of Reviewer 2

Point 1: The most important one is that several experiments have not been described in the Methods section (in particular those reported in paragraph 2.1, 2.2 and 2.3). Therefore this part should be re-written.

Response 1: We feel great thanks for your professional review work on our article. We have re-written “Materials and Methods” according to the Reviewer’s suggestion. In addition, additional methods are described in the Supplementary Materials and Methods.

Point 2: Check for some exponential figure (e.g. 109 page 2 line 83).

Response 2: We feel sorry for our carelessness. We have corrected this mistake, and we have also proof-read and corrected other typos.

Point 3: Please explain NEFA acronym.

Response 3: We have added the full name of NEFA in the revised manuscript.

Best Wishes to you!

Yours sincerely,

Cuiqing Zhao

Round 2

Reviewer 1 Report

When writing the review, specific lines to which the comments relate were indicated. It would be nice of the Authors if they behaved so elegantly and wrote the answer in the same spirit, taking into account the multitude of comments.

Author Response

We would like to express our gratitude for the opportunity to revise our manuscript entitled “Lactiplantibacillus plantarum LPJZ-658 Improves Non-Alcoholic Steatohepatitis by Modulating Bile Acid Metabolism and Gut Microbiota in Mice” (Manuscript ID: ijms-2586433). We sincerely thank all the reviewers for their valuable feedback, which we have carefully considered and used to improve the quality of our work.

We have addressed the reviewers’ comments in a point-by-point manner below, and we have made the necessary revisions to the manuscript. Specific concerns raised by the reviewers have been numbered for clarity. Our responses are given in normal font, and changes/additions to the manuscript are highlighted in red.

Thank you again for your time and consideration.

Response to the comments of Reviewer 1

When writing the review, specific lines to which the comments relate were indicated. It would be nice of the Authors if they behaved so elegantly and wrote the answer in the same spirit, taking into account the multitude of comments.

Point 1: L 23 – unclear

Response 1: We feel sorry for our unclear description. We have made modifications of this sentence in the revised manuscript (P1, Line 21-27).

Point 2: L 31 – unclear

Response 2: We feel sorry for our unclear description. We have made modifications of this sentence in the revised manuscript (P1, Line 33-35).

Point 3: L 44- pathogenesis of what?

Response 3: Thanks for your correction. We have corrected this sentence (P2, Line 47).

Point 4: L 47 – bile acids? It should be clarify

Response 4: We feel sorry for our unclear description. We have clarified it (P2, Line 47).

Point 5: L 55- they are components of bile that is released to intestine

Response 5: Thank you for the question. Yes, bile acids (BAs) are amphipathic molecules that facilitate the absorption of dietary fat and lipophilic vitamins in the small intestine. BAs are synthesised in the liver as primary BAs (cholic acid [CA], chenodeoxycholic acid [CDCA], hyocholic acid [HCA]), secreted in the bile and the intestine where they are transformed by the gut flora into secondary BAs (deoxycholic acid [DCA], lithocholic acid [LCA], hyodeoxycholic acid [HDCA], ursodeoxycholic acid [UDCA]). Each BA species can be measured as free or conjugated to taurine or glycine. In humans, more than 90% of BAs are reabsorbed in the intestine and returned to the liver via the portal vein in an enterohepatic cycle. We have made a short description (P2, Line 51-57).

Point 6: L 67 – if probiotics – are

Response 6: Thanks for your correction. We have corrected this mistake (P2, Line 70).

Point 7: L 73-74- if the two stages are mentioned, the entire process should be described shortly

Response 7: Thank you for your suggestions. We have added brief description in the revised manuscript (P2, Line 73-78).

Point 8: L 77 – this part should be rephrased substantially. There should be presented basal hypothesis and the goal of the study and methods that were used to check the hypothesis

Response 8: We feel great thanks for your professional review work on our article. According to your comments, we have made modifications to our previous description in the revised manuscript (P2, Line 84-95).

Point 9: L 85  - called LPJZ-658

Response 9: Thanks for your correction. We have corrected it (P2, Line 92).

Point 10: L 86 – it is Lactiplantibacillus plantarum

Response 10: Thanks for your correction. We have corrected it (P2, Line 93).

Point 11: L 87 – unclear

Response 11: We feel sorry for our unclear description. We have made modifications of this sentence in the revised manuscript (P4, Line 143).

Point 12: L 89 – what is WD; what is CC14

Response 12: We have added the full name of WD (P3, Line 111) and CCl4 (P3, Line 115) in the revised manuscript.

Point 13: Fig. 1 A should be moved to the material and methods

Response 13: We sincerely appreciate the valuable comments. We have moved Fig. 1A to the material and methods (P3, Line 126-130).

Point 14: L 88- 90 – material and methods

Response 14: We sincerely appreciate the valuable comments. We have moved this part to the material and methods (P3, Line 105-119).

Point 15: L 379-381- unfinished sentence

Response 15: Thanks for your correction. We have corrected this sentence (P3, Line 98-100).

Point 16: part 4.1 – if in L 383 is described the way by which it is given, the dose should be provided with CFU

Response 16: Thanks for your valuable comments. We have corrected it (P3, Line 102-103).

Point 17: L 390 – small letter

Response 17: Thanks for your correction. We have corrected this mistake (P2, Line 112).

Point 18: L 389 – the reference for this Western Diet used needed

Response 18: Thanks for your suggestion. We have cited the references into the revised manuscript (P3, Line 112).

Point 19: L 392- is it tetrachloromethane? It should be clarify

Response 19: We are sorry for our limited description. Yes, the full name of CCl4 is carbon tetrachloride, also known as tetrachloromethane (P3, Line 115-116).

Point 20: L 386- 399- chaotic description. It should be described in line with time

Response 20: Thanks for your valuable comments. We have re-written “Materials and Methods” according to the Reviewer’s suggestion (P3, Line 105-119).

Point 21: L 401 – feces from what part of GIT? And then what?

Response 21: We are sorry for our limited description. Mice cecal contents were collected and stored at -80 ℃, then transported to the laboratory for gut microbiota analysis (P3, Line 121-122).

Point 22: L 402 Liver and serum samples subjected to gut microbiota analysis?

Response 22: We feel sorry for our carelessness. We have corrected this mistake in the revised manuscript (P3, Line 123-125).

Point 23: n of animals in each group should be indicated

Response 23: Thanks for your valuable comments. We have added the description in the revised manuscript (P3, Line 125-126).

Point 24: L 402 – what part of liver was subjected to histological analysis.

Response 24: In this study, we collected the same part of the left liver lobe from mice for histological analysis (Supplementary Materials and Methods P1, Line 21).

Point 25: the description of any methodology is very poor and this way is not accepted

Response 26: We feel great thanks for your professional review work on our article. We have re-written “Materials and Methods” according to the Reviewer’s suggestion (P3, Line 98-131).

Point 26: part of 2.2 the lack of proper histopathology of liver and Figure 2A is very poor and not adequately described.

Response 26: We feel sorry for the unclear histopathology results of Fig 2A. Therefore, we added black arrows to the H&E staining to indicate inflammatory cell infiltration. In addition, we have also described this part of the results in more detail (P5, Line 165-176).

Point 27: L 135 – Sirus Red staining serves to described collagen content but not in light

Response 27: Thanks for your valuable comments. In our study, Sirius Red stain were performed according to the manufacturer’s instructions (Servicebio, Wuhan, China) as described previously 1. In fact, collagen fibers appeared red in under normal light microscopy and other tissue components are stained yellow.

Point 28: Figure 3A does not show collagen

Response 28: Thanks for your valuable comments. Sirius Red is a strong acidic dye, which easily combines with the alkaline groups in the collagen molecule and adsorbs firmly. According to the manufacturer’s instructions (Servicebio, Wuhan, China), collagen fibers appeared red in Sirius Red staining 2.

Point 29: L 137 and fig. 3b – there is lack the description how collagen content was analyzed. The result should be proven

Response 29: We are sorry for our limited description. The percentage of fibrosis area in Sirius Red staining was measured by Image J 1.8.0 (Supplementary Materials and Methods P1, Line 24-25).

Point 30: L 149- there is no inflammation

Response 30: It is possible that the picture of Fig 2A is too small to show clearly. In fact, H&E-stained sections revealed that the liver had inflammatory foci, and we added black arrows to indicate inflammatory cell infiltration.

Best Wishes to you!

Yours sincerely,

Cuiqing Zhao

References

  1. Liang QS, Xie JG, Yu C, et al. Splenectomy improves liver fibrosis via tumor necrosis factor superfamily 14 (LIGHT) through the JNK/TGF-beta1 signaling pathway. Exp Mol Med 2021;53:393-406.
  2. Peng C, Tu G, Yu L, et al. Murine Chronic Pancreatitis Model Induced by Partial Ligation of the Pancreatic Duct Encapsulates the Profile of Macrophage in Human Chronic Pancreatitis. Front Immunol 2022;13:840887.

Reviewer 2 Report

Answers were fine

Author Response

We would like to express our gratitude for the opportunity to revise our manuscript entitled “Lactiplantibacillus plantarum LPJZ-658 Improves Non-Alcoholic Steatohepatitis by Modulating Bile Acid Metabolism and Gut Microbiota in Mice” (Manuscript ID: ijms-2586433). We sincerely thank all the reviewers for their valuable feedback, which we have carefully considered and used to improve the quality of our work.

We have addressed the reviewers’ comments in a point-by-point manner below, and we have made the necessary revisions to the manuscript. Specific concerns raised by the reviewers have been numbered for clarity. Our responses are given in normal font, and changes/additions to the manuscript are highlighted in red.

Thank you again for your time and consideration.

Response to the comments of Reviewer 2

Point 1: Answers were fine.

Response 1: Thank you for your comments on our manuscript.

Best Wishes to you!

Yours sincerely,

Cuiqing Zhao